# South Korea’s National Animal Welfare Policies in Comparison to Legal Frameworks and Systems in Other Countries

**DOI:** 10.3390/ani15091224

**Published:** 2025-04-26

**Authors:** Yeonjin Park, Hochul Shin, Dahee Park

**Affiliations:** Department of Veterinary Pharmacology and Toxicology, College of Veterinary Medicine, Konkuk University, Seoul 05029, Republic of Korea; hshin@konkuk.ac.kr (H.S.); ddaheeya@gmail.com (D.P.)

**Keywords:** animal welfare policy, welfare state policy, South Korea animal welfare

## Abstract

This study analyzes and compares the animal welfare policies of different countries to identify the current status and direction of Korea’s animal welfare policy. It compares objective data on animal welfare laws and systems by sector to examine how animals are recognized as valuable and treated with care in a human-centered welfare state. This study shows that animal welfare policies have varying characteristics depending on the type of welfare state and changes in social dominance and awareness. It suggests that Korea, currently working on enhancing and developing animal welfare policies, requires a paradigm shift in both human and animal welfare.

## 1. Introduction

This study classifies the animal welfare policies of seven welfare states and compares them to analyze qualitative differences across policy types. In doing so, it seeks to determine South Korea’s position within the broader context of animal welfare policies and identify necessary efforts to advance its approach to animal welfare.

This study was motivated by the growing recognition of animal welfare as an alternative framework for addressing social risks in human societies, particularly as welfare policies increasingly shift toward sustainability in response to societal changes. However, despite heightened awareness and the economic significance of animals, the development of animal welfare laws and systems—the foundation of animal welfare policies—has lagged behind.

Until recently, animal welfare was primarily regarded as a moral and philosophical issue, requiring reflection from a human-centered perspective. However, its growing influence on international trade agreements highlights the need for a policy-driven approach. The term *animal welfare* was officially introduced into international trade law through a proposal submitted by the European Union (EU) during the special World Trade Organization (WTO) agricultural negotiations in June 2000 [1]. This proposal, titled *Animal Welfare and Trade in Agriculture*, underscored the EU’s non-trade concerns in the agricultural sector [1]. Many countries subsequently began incorporating animal welfare considerations into laws regulating livestock farming, transportation, slaughter, and the use of animals for scientific purposes. However, in South Korea, animal welfare remained a relatively unfamiliar concept at the time. While trade agreements referenced animal welfare within cooperative frameworks, it is now essential for South Korea to fully embrace the concept and establish comprehensive policy measures.

Current animal welfare policies primarily rely on stringent regulatory frameworks, often resulting in conflicts of interest between those who bear the costs and those who benefit. Without a clear and well-defined rationale for government intervention—specifically addressing why and how the government should act—animal welfare policies risk losing legitimacy and failing to achieve their intended objectives. However, research on human–animal relationships, particularly studies focused on animals, has a relatively short history and remains limited in scope worldwide.

Legal and philosophical debates surrounding animal welfare and rights have gradually gained momentum since approximately 2000. In South Korea, research primarily examined the socio-cultural context of animal rights discourse [2], provided philosophical and legal interpretations of animal rights [3], and analyzed the modern animal rights movement to assess the historical and future trajectories of animal welfare and rights [4]. Although limited in scope, some studies explored changes in the legal status of animals through revisions to animal laws [5] and examined animals’ economic and emotional roles [6,7]. Despite the increasing integration of human–animal interactions within human welfare initiatives, researchers have been hesitant to incorporate animal welfare into this context.

A welfare society refers to government-led initiatives through which the state meets the basic needs of its members and addresses social issues via structured systems and institutions [8]. These needs encompass not only physiological security but also higher-order aspects such as self-actualization, quality of life, and recognition of life’s dignity. Accordingly, advancing animal welfare requires institutional developments driven by government intervention to ensure that animals, as socially vulnerable beings within a human-centric economic society, can coexist with humans as sentient beings. Furthermore, from an ideological perspective, welfare state policies and animal welfare policies share similar objectives.

The type of welfare policies that a society adopts serves as a key indicator of its overall welfare level and quality of life. We contend that this underscores the need for a normative framework that strengthens the legal and institutional foundations necessary for the harmonious coexistence of humans and animals.

South Korea’s animal welfare policies have been largely reactive to date, addressing social issues as they arise rather than following a clear and cohesive vision. In addition, due to inconsistencies and lack of sophistication in policy direction, overlapping or contradictory measures across government departments have been frequently observed, suggesting the need for the systematic reform of laws and institutions. This study seeks to address the limitations of previous research, which primarily focused on animal-related discourse, by examining various policy types that integrate the two core components of the animal welfare policy system: animal welfare laws and animal welfare systems.

Esping-Andersen [9] described typology as a valuable tool for understanding broad policy patterns, facilitating macro-level comparisons. However, typological studies have faced various criticisms, including their inability to adequately explain differences between types, the underlying reasons for such differentiation, and their failure to account for relevant historical and political contexts [10].

This study employed fuzzy set ideal type analysis (FS/ITA) as a typology-building tool. Seven OECD member countries were selected for analysis based on their active implementation of animal welfare policies and alignment with Esping-Andersen’s three welfare state models: Austria, Denmark, Germany, South Korea, Sweden, the United Kingdom (UK), and the United States (USA). The typological framework of animal welfare policies was structured along two axes—animal welfare laws and animal welfare systems—allowing for a comparative analysis of the interaction between regulatory frameworks and implementation across these seven countries. Ultimately, this study sought to determine South Korea’s position within the typology of animal welfare policies and assess which policy approaches could most effectively advance animal welfare in the future.

### 1.1. Theoretical Background

#### Human Welfare and Animal Welfare

The establishment of animal-related laws and systems in human societies dates to 1822 with the enactment of Martin’s Act in England. Named after Richard Martin, who advocated for legislation to prevent animal abuse, Martin’s Act was the first animal protection law aimed at preventing cruelty and the unfair treatment of animals [11]. Notably, the origins of England’s child protection laws can also be traced to efforts to safeguard both animals and children. This shared foundation links the Royal Society for the Prevention of Cruelty to Animals (RSPCA) and the National Society for the Prevention of Cruelty to Children (NSPCC) [12]. Founded in 1824, the RSPCA supported causes such as the abolition of slavery and child protection, reflecting broader concerns for societal welfare.

Early English laws primarily focused on the treatment of the working class and animals. However, after World War I, welfare priorities shifted predominantly to humans, influenced by advancements in sociology and psychiatry. This shift led to the exclusion of animals from welfare considerations, shaping modern welfare policies with a strong anthropocentric bias. In the rare instances where animals were included, it was typically for their utility to humans. Ryan [13] argued that this prevailing anthropocentrism in the welfare sector has hindered a more comprehensive understanding of *humans as animals*.

Since the 1990s, animal welfare has increasingly been emphasized in international discussions as an ethical concept separate from the nature of the relationship between humans and animals. These arguments challenge us to reconsider the essence of humanity and the scope of our moral responsibilities. Just as human welfare has advocated for vulnerable members of society and emphasized morality, it is our duty to represent and assign moral standing to animals—the most vulnerable members of our ecosystem. Incorporating animals into human welfare remains a relatively new challenge compared to other fields. We posit that it is now time to clearly recognize the zoological connection to human welfare and initiate behavioral changes not just in theory but also in practice. In the 21st century, discussions abroad have increasingly emphasized that animal welfare is an ethical concept, regardless of the nature of human–animal relationships. These arguments challenge us to reconsider the essence of humanity and the scope of our responsibilities. Just as human welfare has advocated for vulnerable members of society and emphasized morality, we have an obligation to represent and assign moral status to one of the most vulnerable groups—animals. In fact, incorporating animals into human welfare remains in its early stages compared to other fields. It is time to recognize the zoological connections to human welfare and implement changes in theory and in practice.

Those who hold racist ideologies and wield power over others often depict their victims as lacking value and sensitivity [14]. This pattern aligns with historical instances of human slavery, genocide, and the exploitation of women [15]. Recognizing the intersections between human and animal welfare requires greater attention to how we treat animals [16]. To this end, inflicting unnecessary harm on animals should be avoided, and animal welfare must be considered in industries involved in the mass killing of animals.

Calvo [17] argued that institutionalized animal exploitation for meat production is shaped by the “relationship between capital and patriarchy”, primarily through the breeding of female animals to produce “protein”. This includes the consumption of milk from cows and the forced breeding of female animals to maximize profits. Feminist scholars contend that such practices reinforce patriarchal systems over time [18].

The above studies suggest that a critical examination of speciesism, sexism, and power structures should accompany discussions of animal welfare to provide deeper insights into human welfare.

### 1.2. The Concept of Animal Welfare Policy

Animal welfare policies have evolved alongside changes in animal-related laws and systems, shaped by the shifting societal perceptions of animals. Just as past policies have been grounded in specific values, animal welfare policies are rooted in the fundamental ideology of *animal welfare.* If animal welfare is defined as a “state in which an animal’s basic needs are met and its suffering is minimized” [19], then all actions promoting humane treatment can be considered part of animal welfare policies.

The formation of animal welfare policies has consistently been influenced by the priorities of major economic actors, with economic viability remaining a key concern. Amid conflicting interests among stakeholders, these policies primarily focused on aspects of animal protection, leaving the broader concept of animal welfare largely unaddressed. However, as countries increasingly adopt more progressive perspectives on animals, the perception of animals as mere objects has shifted toward recognizing their intrinsic value and rights. Some nations have even enshrined the dignity of animal life within their constitutions. Consequently, animal welfare policy can be redefined as “laws and systems that ensure the basic needs of animals as living beings, minimize their suffering, and promote their well-being”. This shift reflects a growing economic, social, and political interest in animals, which were previously considered primarily from philosophical and ethical perspectives, and underscores the need for more comprehensive legal and administrative approaches.

Extensive research across various fields examined the laws and systems governing human welfare, aiming to sustain and enhance the prosperity of human society. However, recent trends—including ongoing social and environmental changes, the expansion of policy subjects, and uncertainties surrounding intrinsic value—have emerged as direct drivers of policy transformation. Alongside anthropocentric policy frameworks, there is a growing argument—voiced primarily by animal advocacy groups and increasingly echoed in political circles—that animal issues should be addressed from multiple perspectives, emphasizing shared fundamental values. The rising public interest in animals and increasing awareness of animal welfare have brought animal welfare policies to the forefront of social discourse.

### 1.3. Distinctiveness from Previous Studies on Animals

This study analyzed animal welfare policies through the lens of welfare state typologies, offering four key distinctions from prior research.

First, it critically revised mainstream welfare state typologies to provide a macro-level perspective on animal welfare policies. Specifically, it introduces a new analytical framework from the perspective of an animal welfare researcher, building upon Esping-Andersen’s welfare state model and gender typologies, which focus on social class relationships. While it does not propose a fully developed typology theory for animal welfare policies, it demonstrates the potential for theory building based on the characteristics of animal welfare policies and highlights the need to examine them alongside human welfare.

Second, given the scarcity of animal-related research in the social sciences globally, this study adopted an integrative and systematic approach to measuring animal welfare policies across countries. Previous national typology studies in other disciplines often relied on overly theoretical or ideological approaches that lacked practical applicability for categorizing countries and failed to account for the unique environmental characteristics of each nation. In contrast, this study emphasizes practicality by conducting detailed measurements across various domains within each country and developing inductive typologies based on empirical data.

Third, while previous studies on animals primarily focused on conceptual debates or conducted limited comparative analyses centered on specific animal welfare issues, this study evaluated animal welfare policies across multiple countries from both micro- and macro-perspectives. Additionally, it examined South Korea’s current position and outlined future policy directions within this framework.

Lastly, this study is the first of its kind, both domestically and internationally, to attempt a typology of animal welfare policies. Pertinent characteristics were identified through a comprehensive examination of the laws and systems governing animal welfare from a policy perspective. By using empirical data as the unit of analysis, this study minimized the abstractness that often arises in qualitative research.

## 2. Materials and Methods

### 2.1. Research Model

A country’s animal welfare policies establish goals and concrete measures systematically implemented through animal welfare laws and systems. These laws safeguard animals as socially vulnerable beings, mediate social and economic conflicts, and account for animals’ natural lives. In this respect, they align closely with welfare state principles, with their emphasis on welfare rather than mere protection adding significant value. However, effective enforcement requires appropriate tools and resources, regardless of how well-crafted the legal framework may be.

This study selected seven countries from the 37 OECD member states with animal welfare legislation, applying the welfare state typology introduced in Esping-Andersen’s [8] *The Three Worlds of Welfare Capitalism*, which categorizes welfare states into three types: liberal, conservative, and social democratic. The objective was to classify these countries based on the characteristics of their animal welfare policies and identify key differences between them. To conduct the typology analysis, this study employed the FS/ITA method, which is particularly well suited for research topics that cannot be easily expressed through linear relationships. This method allows for identifying the degree to which cases—beyond simple dichotomies—belong to a specific type and the extent to which they exhibit other attributes, thereby enhancing our understanding of their characteristics. Through this approach, this study derived theoretical and policy implications for developing an optimal animal welfare policy model for South Korea. The research model is presented in Figure 1.

### 2.2. Selection of Case Countries and Data

The countries selected for comparison through FS/ITA were chosen from the 35 OECD member states with animal welfare laws (from 37 OECD member states, as of June 2018). The selection criteria were as follows:(i)Countries with relevant provisions in their constitutional laws, animal protection laws, or animal welfare laws;(ii)Countries whose constitutional or civil law provisions grant animals a distinct legal status;(iii)Countries with relevant provisions in local laws [20,21].

The concepts used as units of analysis for constructing the hypotheses had to be empirically verifiable. Laws and systems, as tangible constructs broadly definable through consensus, are particularly well suited for translating abstract concepts into empirical data for theory building. Consequently, *the social value of animal welfare laws* and *animal welfare systems*, which hold universal relevance across diverse social and political contexts, were chosen as key indicators. To examine the legal and value-based dimensions of animal welfare laws more systematically, we developed an objective scoring system to quantify this otherwise difficult-to-measure concept. Using the criteria derived from these two analytical indicators, we refined the selection of countries for comparison.

As mentioned, 20 of the 35 countries initially examined were selected based on their alignment with the measurement criteria for the four dimensions of animal welfare systems: responsibility, resource allocation, committees, and education. This selection process involved reviewing national websites to assess the presence or absence of information pertaining to these dimensions.

Seven countries were ultimately chosen for comparative analysis based on the following inclusion criteria:(1)The analysis period was set from 2010 to 2018 to account for variations in the timing of the establishment of animal welfare laws and systems across countries.(2)Countries representing Esping-Andersen’s three welfare models were included—namely, Austria, Denmark, Germany, Sweden, the UK, and the USA—which are relatively well known in South Korea. South Korea was also included to facilitate specific comparisons with these countries and derive relevant policy implications.

The primary reason for limiting the number of case countries was to ensure stable access to data. Selecting an equal number of countries from each welfare model helped minimize subjective and relative factors that could influence the standardization process used to generate ideal-type membership scores.

### 2.3. Analysis Methods

#### 2.3.1. Fuzzy Set Ideal Type Analysis (FS/ITA)

Fuzzy set ideal type analysis calculates membership scores for comparative cases based on specific attributes and ratings, allowing for an analysis of the extent to which each case aligns with a specific type [22,23]. Following the fuzzy set ideal type analysis procedure outlined by Kvist [24], FS/ITA in this study was conducted as follows.

First, ideal types were defined to construct a property space, incorporating both theoretical and empirical knowledge. Second, fuzzy membership scores were calculated for each case within this property space. Third, the number of ideal type models was determined by 2^n^, where *n* represents the number of indicators. In this study, four ideal type models were generated.

The fuzzy membership scores for each model were calculated using the principle of complementarity and the minimum operator [23]. The principle of complementarity refers to the complement of a set, denoted as 1-A and represented by the value *a*. When combining two sets, the minimum operator was used, assigning the lowest score as the fuzzy membership score for each case. Conversely, in the union of two sets, the maximum operator is applied, selecting the highest fuzzy membership score for each case. After completing these steps, the fit of each case to the ideal type models was evaluated.

#### 2.3.2. Selection of Indicators and Formation of the Property Space

Animal laws currently lack a standardized classification framework and can therefore be categorized from various perspectives. This study adopted a classification approach that emphasizes legal categories commonly addressed within administrative systems and widely recognized in society, focusing on the individuality of animals. Social value is typically defined as *values that contribute to public interest and community development.* However, for social value to be realized as a legal concept, it must first be granted legal significance by the constitution.

For instance, under the principles of the Korean Constitution, social value aligns ideologically with the welfare state principle. This principle not only ensures socio-economic security for vulnerable individuals but also seeks to address conflicts and inequalities between social classes, using policies to establish a fair and just social order. In this context, examining the social value of animal welfare laws offers a more advanced and holistic approach to understanding the lives of animals.

Animal welfare systems encompass the entire policy process, from decision making to implementation, involving both internal administrative management and the participation of various external stakeholders. To develop analytical indicators for animal welfare systems, this study drew on the social welfare policy analysis framework proposed by Gilbert and Terrell [25]. Using these two indicators—*Social Value of Animal Welfare Laws* and *Animal Welfare Systems*—a four-quadrant property space was constructed, as illustrated in Figure 2.

#### 2.3.3. Dimensions of Each Indicator


Social Value of Animal Welfare Laws


To assess the social value of animal welfare laws, common categories such as companion animals, farm animals, laboratory animals, exhibition animals, and wild animals were selected as key dimensions, as shown in Table 1. The measurement items for each dimension include global issues focusing on the prevention of animal abuse and the provision of adequate living conditions for animals.


2.Animal Welfare Systems


The dimensions for the Animal Welfare Systems indicator were identified by prioritizing key policy issues, as shown in Table 2, and include responsibility, education, committees, and resource allocation.

## 3. Results

### 3.1. Derivation of Types

This study utilized the indicators of animal welfare law–social value and animal welfare system–support for fuzzy set ideal type analysis, grouping cases based on their membership degree in the predetermined fuzzy sets. In this process, qualitative case analysis was conducted not at the researcher’s discretion but based on existing theoretical and substantive knowledge [22].

Before converting the measurement values of each dimension into fuzzy scores, several factors needed to be considered. Since each indicator consisted of multiple dimensions, it was necessary to determine whether these dimensions had equal influence on the indicator and whether their measurement values were evenly distributed. In this study, the dimensions were found to have slightly varying influences on the indicator. Therefore, after repeatedly comparing the values of the dimensions, empirical scores were calculated. Each dimension was assigned a total score of 10 points, divided into two measurement items, with a maximum of 5 points per item. The measurement criteria for these items were maintained at similar intervals. However, while other dimensions had two measurement items (each with a maximum of 5 points), the *animal legislation type* dimension only consisted of only one measurement item, creating an imbalance in weighting among dimensions. To address this, a weight of 2 points—the lowest score— was assigned to all measurement subitems within the *animal legislation type* dimension.

The measured scores were converted into fuzzy scores using the measurement functions of the fs/QCA 2.0 program. These fuzzy scores were then averaged to generate fuzzy membership scores for the indicators. The full membership value was set to the maximum empirical score, the cross-over point to the median empirical score, and the non-membership value to the minimum empirical score. The degree of full membership was set at 0.95, while the degree of non-membership was set at 0.05.

Based on this process, the fuzzy membership scores for *animal welfare law—social value* were calculated, as shown in Table 3.

Table 4 presents the fuzzy membership scores assigned to the indicator “animal welfare system—support” calculated for the overall indicator and its dimensions.

Based on the fuzzy membership scores calculated above, the fuzzy intersection function and the principle of negation were applied to compute the fuzzy membership scores for each type across all case countries (Table 5). The FS/ITA results for the seven case countries determined their respective ideal types. Type AB included the UK (0.6500), Denmark (0.6125), Germany (0.5450), and Austria (0.8375), whereas type ab included the USA (0.7925), Sweden (0.5250), and Korea (0.9325).

### 3.2. Typology Analysis

While the interpretation of the ideal type focused on the conceptual meaning of the property space, the interpretation based on actual types classified cases according to their degree of membership within each type. For example, a comparison of the fuzzy membership scores for the four countries classified as *AB types* in Table 5 reveals that, although they share the same classification, their laws and institutions differ depending on their degree of membership.

Austria, with a membership score of 0.8375, and the UK, with a score of 0.6500, exhibit higher degrees of membership compared to Denmark (0.6125) and Germany (0.5450). The relatively lower scores of Denmark and Germany within the *AB type* suggest the possibility of partial alignment with another type. Indeed, both countries demonstrated slightly higher levels of animal welfare system support and social value of animal welfare laws relative to the benchmark. In such cases, fuzzy set analysis allows these countries to be interpreted as *AB types with some characteristics of type aB.*

Although the UK also demonstrated moderate membership in the *AB type* (0.6500) similar to Denmark and Germany, its membership score for the system support indicator was significantly higher at 0.8050, compared to Denmark (0.6125) and Germany (0.5450).

An analysis of the seven countries identified Austria as the representative *AB type* and the USA and South Korea as representative *ab types.* Denmark and Germany were classified as *AB types with some characteristics of type aB,* while Sweden was categorized as an *ab type with some characteristics of type aB.*

A closer examination suggests that Denmark and Germany may align more closely with a different type, sharing some characteristics with Sweden (*aB type*). However, all three countries exhibited their respective type characteristics to varying degrees. Positioned near the threshold, they balanced attributes of both *AB* and *ab* types but remained further from the representative *AB type* (Austria) and *ab type* (the USA and South Korea). Essentially, while Denmark, Germany, and Sweden were formally classified as part of both *AB* and *ab* types, they appeared to converge into a distinct type that uniformly embodied *aB type* characteristics.

Figure 3 provides a comprehensive overview of the four actual types, the respective countries, and their defining characteristics, as derived from in-depth analyses and interpretations. Detailed explanations of these four types are presented in the following subsections.

#### 3.2.1. Economic Value Type

While the USA established its Animal Welfare Act and institutional foundations earlier than South Korea, the membership scores for the two countries showed little difference. This may be attributed to the fact that the measurement items primarily assessed laws and systems at the federal/central government level, excluding significant animal welfare laws and systems implemented at the state level in the USA. Although the USA exhibited some shortcomings in both the social value of welfare laws and system support in theoretical and empirical analyses, the validity of the results remains intact, as the same measurement criteria were applied to other federal states.

In South Korea, most welfare needs were traditionally met by the private sector rather than the government, reflecting the country’s free-market economic system. However, following rapid economic growth, the financial crisis of the late 1990s prompted a shift toward universal welfare, breaking away from the path dependency typical of general welfare systems. This shift began to influence South Korea’s animal welfare policies, gradually incorporating elements of universal welfare—welfare benefits accessible to all citizens regardless of conditions or qualifications. This transition introduced a new paradigm in perceiving animals, shifting from a market-driven perspective to one emphasizing *harmonious coexistence between humans and animals* [26].

Although South Korea is classified as a neoliberal economy, it exhibits characteristics of a welfare mix, particularly in its approach to animal welfare. This is reflected in relevant laws and systems that prioritize the humane treatment of animals.

Both case countries share a common foundation in capitalism and exhibit characteristics of a liberal welfare model. Their animal protection and welfare laws primarily focus on preventing animal abuse. Given their policy emphasis on private sector involvement rather than government regulation, the legal enforcement of animal cruelty prevention measures has been relatively weak compared to other countries. Regarding animal welfare system support, both countries widely recognize the private ownership of animals and prioritize economic efficiency in their treatment. Government support has been largely supplementary, serving as a passive mediator among interest groups rather than an active enforcer.

Thus, compared to other types, this model reflects a hierarchical and instrumental view of animals under industrial capitalism, driven by human needs. It was therefore labeled the *economic value type.*

#### 3.2.2. Social Value Type

The UK was the first country in the world to establish animal welfare standards and legislate against animal cruelty. Even before the EU introduced similar measures, the UK had already implemented laws and systems prohibiting activities that violated animal welfare standards, demonstrating proactive administrative efforts in this area.

The UK government’s policy of sharing costs and responsibilities with the animal industry to reduce the burden of legal and regulatory compliance is widely regarded as commendable. However, its effectiveness has yet to be fully verified. Given past financial pressures from diminishing government funding, doubts remain about whether the government can sustain the financial burden of regulating the animal industry in the long term.

While the UK shares the liberal welfare model characteristics of *ab-type* countries such as the USA and South Korea, it diverges significantly in its approach to animal welfare policies. This divergence is largely due to the extensive lobbying efforts of the RSPCA and strong public support for animal welfare, both of which have played a crucial role in driving the UK government’s progress in this field.

This type was classified as the *social value type* as it represents a capitalist economy that promotes animal welfare by enhancing the efficiency of social systems and prioritizing animal well-being.

#### 3.2.3. Rights Expansion Type

Sweden and Denmark, recognized as social democratic welfare states, and Germany, classified as a conservative welfare state, share *aB-type* attributes, where the social value of animal welfare laws slightly exceeds the level of support provided by animal welfare systems. These countries explicitly designate the protection of natural environments and animals as state responsibilities in their constitutions or animal welfare laws. Moreover, their legislation includes detailed guidelines and implementation standards to ensure that animal welfare policies operate within a clear and structured framework.

These *aB-type* countries recognize the inherent, inalienable value of animals and adopt proactive measures to improve their living environments and alleviate suffering. All three countries strictly comply with the EU’s stringent animal welfare standards.

A defining characteristic of these countries is the widespread social recognition of animals as sentient beings, which has facilitated more active implementation of animal welfare policies. This type was therefore classified as the *rights expansion type.*

#### 3.2.4. Ecological Type

Austria, which fully aligns with the *AB type*, explicitly enshrines animal protection in its constitution and has established a robust administrative system to emphasize animal welfare as a national priority. For instance, every political party in Austria appoints animal welfare advocates, and each state has an animal welfare ombudsman who serves as a non-governmental representative, advocating for animal welfare and maintaining close communication with state governments. Austria’s animal welfare enforcement system relies on evidence-based scientific data and engages in extensive stakeholder consultations to continuously improve policy quality.

The principle of coexistence is deeply ingrained in Austria’s governance approach. The country strongly integrates economic development with environmental protection, firmly rejecting the notion that ecological concerns hinder growth. This perspective has enabled Austria to expand its socio-economic activities in an environmentally sustainable manner.

This type was aptly labeled the *ecological type* as it prioritizes harmony between human and animal welfare within social, cultural, and environmental contexts, striving for a sustainable society that promotes the healthy coexistence of humans and animals.

## 4. Discussion

### 4.1. Differences by Type

As society’s perception of animals lends legitimacy to advocacy for animal welfare policies and influences the choice of policy instruments, it is crucial to define the characteristics of animals—the target of such policies—to ensure successful implementation. This study examined two key factors contributing to the emergence of various animal welfare policy types.

First, with the rise in social hierarchy and domination [27], nature came to be viewed as separate from the human community, a subordinate entity subject to human control. The second factor considered was that changes in social perception were deemed more important than objective reality for the policy’s target group.

According to Schneider and Ingram’s [28] theory of social construction, policies can vary depending on how people perceive and treat specific groups. This theory assumes that entities do not exist independently of human perception but are instead shaped by how individuals interpret and understand them. Therefore, analyzing these perceptions is essential. Consideration of these two factors can effectively explain the differences in animal welfare policies across various contexts.

First, the USA and South Korea, categorized under the *economic value type*, exemplify the complete domination of animals within capitalist economies. In these societies, profit-driven capitalism reinforces hierarchical social structures, emphasizing private ownership and human superiority over animals. Animals are treated as commodities, exchanged between producers and consumers in a free market. Ultimately, all entities, including animals, are reduced to exchangeable objects, and society itself functions as a vast production system.

Second, the UK, representing the *social value type*, demonstrates the substantial influence of civil society organizations advocating for animal welfare. These organizations play a pivotal role in shaping state policies, with the government acting as a mediator, balancing the demands of these groups with economic interests.

Third, the *rights expansion type*, represented by Denmark, Germany, and Sweden, acknowledges animals’ autonomy even within systems that reinforce human dominance. These countries have developed rigorous frameworks to implement animal welfare policies that closely resemble the recognition of animal rights.

Fourth, the *ecological type*, represented by Austria, views animals as diverse species that evolve in symbiotic complementarity with one another. This type rejects the notion that animals and humans exist in a world governed by the survival of the fittest. Instead, all species, including humans, are seen as autonomous entities engaged in a symbiotic coexistence characterized by mutual influence and interdependence. In such a society, the hierarchy between humans and animals is dismantled, and animal welfare is recognized as a policy priority.

In summary, variations in societal perceptions of animals and the structures of social hierarchy and domination help explain differences in animal welfare policies. Historically, animals have been considered unworthy of political recognition within human societies. However, societal perceptions are gradually shifting, replacing negative images with more positive ones that acknowledge animals as beings in need of care and support. This evolving perspective is expected to positively influence and refine the direction of animal welfare policy design in the future.

### 4.2. Implications and Recommendations

This study analyzed and classified animal welfare policies in welfare states to compare their differences and derive implications for South Korea. The key findings and recommendations from the analysis are summarized below.

First, the scope of the welfare concept must be expanded. As family structures and industrial landscapes evolve, many welfare states have begun exploring transformations, increasingly recognizing animals as an integral part of human welfare. For instance, Austria pursues an eco-centric society grounded in a market economy and social justice, and some researchers have advocated for a paradigm shift in the concept of human welfare [29,30].

Second, this study provides a multidimensional understanding of South Korea’s animal welfare system through an in-depth comparison with other countries. However, while other countries have developed their animal welfare policies over an extended period, South Korea’s policies remain largely fragmented and reactive to emerging issues. Therefore, rather than directly adopting ideal animal welfare policies from other welfare states, efforts should focus on identifying tailored solutions that acknowledge the gap between the best international practices and Korea’s current reality. This approach could help minimize inefficiencies and prevent resource wastage caused by overlapping or redundant policies.

This study is significant as the first to quantify and classify animal welfare policies in welfare states, providing an in-depth analysis of various legal and systemic dimensions. However, three limitations should be noted, along with recommendations for addressing them:Lack of a comprehensive database on animal welfare laws and systems: Differences in administrative systems across countries made data collection challenging, requiring significant effort to obtain relevant information for each unit of analysis. To address this, an integrated information-sharing system should be developed to manage data by animal classification at both national and international levels.Absence of standardized legal terminology: Appropriate legal terminology specific to animal welfare has yet to be established, which can lead to inconsistencies in policy interpretation and implementation.Limited exploration of influencing factors and policy evolution: While this study successfully classified animal welfare policies and identified policy characteristics in the case countries, it did not examine the factors shaping these policies, their historical development, or potential future trends within each type. Future research should focus on these aspects to provide a more comprehensive understanding.

## 5. Conclusions

There is currently unprecedented interest in and demand for amendments to animal laws in South Korea. This study revealed that, while South Korea follows the trajectory of a liberal welfare state, its animal welfare policy reforms have the potential to evolve into a new and distinctive model. By analyzing the strengths and weaknesses of animal welfare policies in other countries and incorporating these insights, South Korea could establish a strong theoretical foundation and set a clear course for the future development of its animal welfare policies.

## Figures and Tables

**Figure 1 animals-15-01224-f001:**
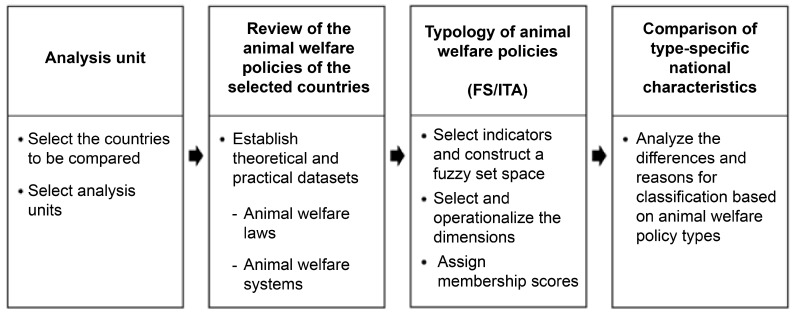
Research model.

**Figure 2 animals-15-01224-f002:**
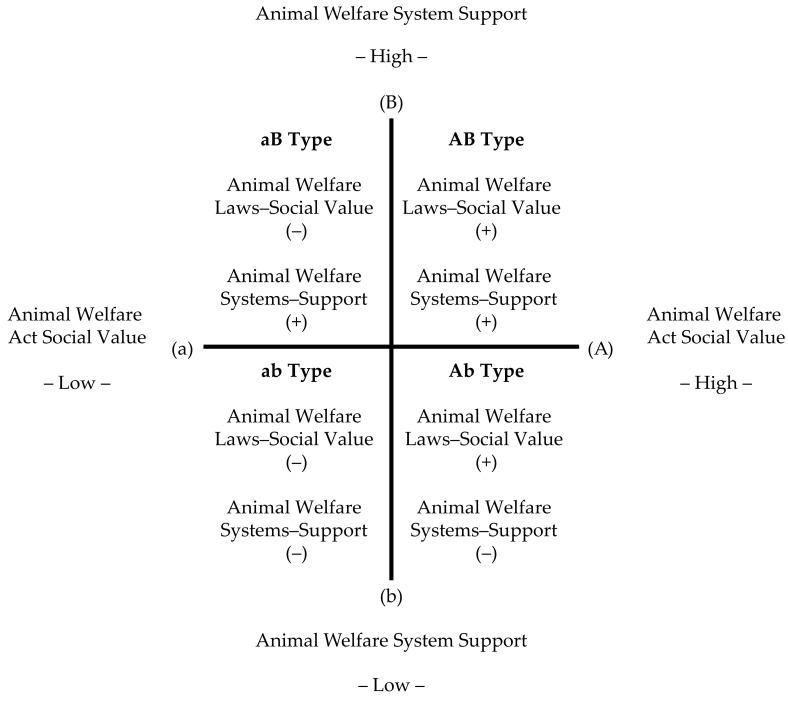
Construction of a classification framework.

**Figure 3 animals-15-01224-f003:**
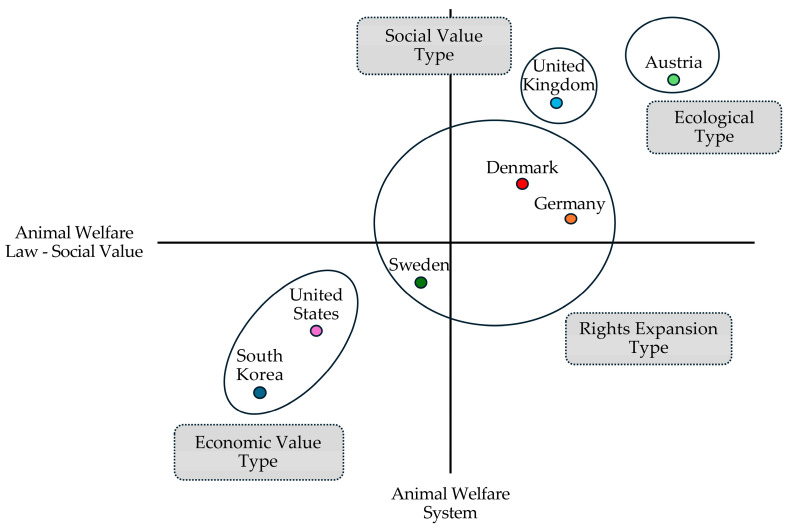
Typology-based classification of the animal welfare policies across countries.

**Table 1 animals-15-01224-t001:** Dimensions and measurement items for evaluating the social value of animal welfare policies.

Indicator	Dimension	Measurement Items
Animal Welfare Laws–Social Value	Animal Legislation Type	Constitution (basic law), animal welfare law
	Farm Animal Law	Humane slaughter
		Legal enforcement of welfare standards for farm animals
	Companion Animal Law	Licensing system for companion animal supply
		Legal enforcement of welfare standards for companion animals
	Laboratory Animal Law	Strict adherence to the 3r principles
		Legal enforcement of welfare standards for laboratory animals
	Wildlife Law	Compliance with the CITES agreement
		Inclusion of animal welfare provisions in hunting laws
	Exhibition Animal Law	Minimization of entertainment factors
		Legal enforcement of welfare standards for exhibition animals

**Table 2 animals-15-01224-t002:** Dimensions and measurement items for evaluating the social value of animal welfare systems.

Indicator	Dimension	Measurement Items
Animal Welfare Systems—Support	Responsibility	Specification of animal welfare responsibilities within government departments
		Development of animal welfare plans and publication of reports
	Resource Allocation	Interdepartmental discussions on animal welfare
		Government budget allocation for animal welfare
	Committees	Diversity of committees under government departments
		Policy authority of committees under government departments
	Education	Inclusion of animal welfare responsibilities in educational objectives
		Integration of animal welfare education into regular preschool and elementary school curricula

**Table 3 animals-15-01224-t003:** Fuzzy membership scores assigned to the indicator “animal welfare law—social value”.

Country	Fuzzy Membership Scores for Individual Dimensions						Overall Fuzzy Membership Score
	Animal Welfare Legislation	Farm Animals	Companion Animals	Lab Animals	Wildlife	Exhibition Animals	
USA	0.50	0.18	0.05	0.05	0.05	0.05	0.1466
UK	0.50	0.95	0.50	0.50	0.95	0.50	0.6500
Sweden	0.50	0.50	0.35	0.50	0.50	0.50	0.4750
Denmark	0.50	0.95	0.73	0.50	0.50	0.50	0.6133
Germany	0.95	0.50	0.73	0.73	0.50	0.65	0.6766
Austria	0.95	0.50	0.95	0.95	0.95	0.95	0.8750
South Korea	0.05	0.05	0.14	0.05	0.05	0.05	0.0650

**Table 4 animals-15-01224-t004:** Fuzzy membership scores assigned to the indicator “animal welfare system—support”.

Country	Fuzzy Membership Scores Assigned to Each Dimension				Overall Fuzzy Membership Score
	Responsibility	Resource Allocation	Committee	Education	
USA	0.50	0.05	0.23	0.05	0.2075
UK	0.50	0.95	0.95	0.82	0.8050
Sweden	0.50	0.73	0.14	0.50	0.4675
Denmark	0.95	0.50	0.50	0.50	0.6125
Germany	0.50	0.50	0.50	0.68	0.5450
Austria	0.50	0.95	0.95	0.95	0.8375
South Korea	0.05	0.12	0.05	0.05	0.0675

**Table 5 animals-15-01224-t005:** Dominant type of each country analyzed.

Country	AB Type	Ab Type	aB Type	ab Type	Dominant Type
USA	0.1466	0.2075	0.1466	**0.7925**	ab
UK	**0.6500**	0.3500	0.1950	0.1950	AB
Sweden	0.4675	0.4675	0.4750	**0.5250**	ab
Denmark	**0.6125**	0.3867	0.3875	0.3867	AB
Germany	**0.5450**	0.3234	0.4550	0.3234	AB
Austria	**0.8375**	0.1250	0.1625	0.1250	AB
South Korea	0.0650	0.0675	0.0650	**0.9325**	ab

A: high support for animal welfare systems, a: low support for animal welfare systems, B: high social value of animal welfare laws, b: low social value of animal welfare laws; and bold values indicate the representative figures that determine the type.

## Data Availability

Data presented in this study are available upon request from the corresponding author.

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
