# Peer review of "South Korea’s National Animal Welfare Policies in Comparison to Legal Frameworks and Systems in Other Countries"

_animals, 2025, doi:10.3390/ani15091224_

Round 1
Reviewer 1 Report
Comments and Suggestions for Authors
This study is an interesting contribution to the field of animal welfare legislation and policies. My main concern is that the paper seem to stand alone and miss the broad qualitative literature on these aspects. I recommend the authors to strengthen the paper by consulting this kind of literature to better frame the issue. As I will show below, some of the statements are not in line with the common view within Europe and North America. I am totally fine with the fact that you might reach another conclusion than the common view, but you need to discuss and problematize the common view.
Line 51-53: I disagree with the authors' statement for the countries in the study. Several countries in Europe had already animal welfare legislation at least in parts by 2000.
Line 64-65: I disagree with the authors' statement. Animal well-being (=welfare) was discussed heavily in the 1960s in Britain as well as animal suffering in the 1970s-1980s. This is well described in Mike Radfords book on Animal Welfare Law in Britain.
Line 115-120: These statements need a proper reference.
Line 128-129: "In recent years". In Europe and the US the discussion has been ongoing since early 1990s with two big conferences in 1993 and 1996 trying to establish and define animal welfare both as a scientific and ethical concept.
Line 163-165: You have an interesting choice of definition for animal welfare and you might to justify it. In Europe and North America this definition could maybe be used but is not among the common ones in animal welfare science (cf. Lerner 2008 (see comment on references above)).
Line 168-169: I guess that you refer to the South Korean setting. Animal welfare has been evident in Europe since the 1990s.
Line 298-299: That animal laws lack a standardized classification framework depends on what you intend to say. For example one can differ between civil law and common law systems (which are both included in this study). Raffael N Fasel & Sean C Butler 2023 Animal Rights Law, Hart Publishing is one recent attempt to make a standardized classification framework.
Line 331: Check the numbering.
Line 359: What is subitems? Are they presented somewhere?
Line 390-399: I get confused by the presentation here because you use results from more than table 5. Please add which tables you get your numbers from for better clarity. What level of significance are you talking about in line 399.
Section 3.2.1-3.2.4: Note that you are not fully consistent between the four categories. You seem to mention different things for different categories. Think about if you could add things on all aspects in all categories.
Line 480-481: This is an interesting conclusion. There has been a long on-going conflict on the opposite, that Denmark (and other countries) avoids compliance with the EU directive on pigs, see Lerner & Algers 2013 The ethics of consumption: The citizen, the market, and the law | SpringerLink
Discussion: As mentioned in the passage on references above. I lack a discussion on other sources analyzing similar aspects of development of legal frameworks and animal welfare policies.
Reviewer 2 Report
Comments and Suggestions for Authors
It was a pleasure to read/evaluate this manuscript. It is a fundamentally a very good paper that most definitely deserves to be published.
The overall approach of this investigation, as well as the structure of this manuscript, represent sound reasoning and careful analysis. In all, this is an example of excellent scholarship.
Having said that, I offer the following editorial comments (nearly all of which pertain to the Introductory portions of this paper):
Line 74: A reference is needed here. Or a statement that the authors are providing the formal definition of “welfare society”.
Line 87: A reference it needed here. Or an indication that this is the authors’ opinion (that could be something like “It could be argued that this underscores……”)
Line 89-90: Same comment.
Line 109: “could most effectively advance animal welfare in the future in South Korea.”
Line 112-119: A reference is needed here.
Line 121-125: A reference is needed here.
Line 128-134: References are needed here.
Line 134-144: It should be made clear that this is opinion. Something like: “The opinion is offered here that it is time to clearly recognize …….”
Line 145-158: Interesting points are made here that do deserve thought. But placed here, they divert from the line of reasoning in this paper. Furthermore, the authors do not return to these themes in their analyses or in their discussion. In this paper, this portion strikes me as non sequiturs. I recommend deleting Lines 145-158 as not relevant to the present paper.
Line 179-180: A reference is needed here.
Line :219-227: This paragraph belongs in the Introduction, not in Methods.
Lastly, I wonder if the authors would consider an alternative Title for this paper. After all, it is clearly designed to evaluate the practices in South Korea in comparison to other nations, and I think it would be better if the title reflected that. I offer something like the following as perhaps more directly descriptive: The National Animal Welfare Policies of South Korea in Comparison to the Legal Frameworks and Systems in Other Countries
In closing I want to emphasize that these suggestions are offered in the hope of being helpful. This is already a very fine paper, and the authors are to be congratulated for bringing it to fruition.
Author Response
Response to Reviewer’s Comments
Dear Reviewer #2,
We sincerely thank you for your thoughtful suggestions and insights that have enriched the manuscript and produced a better and more balanced account of the research.
2 Reviewer(s)' Comments to Author:
- Line 74: A reference is needed here. Or a statement that the authors are providing the formal definition of “welfare society”.
Reply:
We have added a reference as per your suggestion (Line 74).
- Line 87: A reference it needed here. Or an indication that this is the authors’ opinion (that could be something like “It could be argued that this underscores……”)
Reply:
The sentence in question reflects our opinion based on our experience with animal welfare policies being implemented in Korean society. We have amended it accordingly (lines 87–88).
- Line 89-90: Same comment.
Reply:
Lines 89-91 are the author's experience in animal welfare policy research and animal welfare projects.
- Line 109: “could most effectively advance animal welfare in the future in South Korea.”
Reply:
This paper focused on categorizing the animal welfare policies of welfare states, examining the characteristics of each type, and understanding the characteristics of Korea's animal welfare policy. In future research, we plan to compare the time-series flow of animal welfare policies by country and discuss ways to effectively develop Korea's animal welfare policy. We have amended the sentence according to your suggestion (lines 107–110).
- Line 112-119: A reference is needed here.
Reply:
We have added relevant references as suggested (lines 113–122).
- Line 121-125: A reference is needed here.
Reply:
References:
Early English laws primarily addressed the treatment of the working class and animals [11].
Over time, welfare priorities shifted predominantly to humans, influenced by advancements in sociology and psychiatry [8].
- Line 128-134: References are needed here.
Reply:
These statements reflect our research-based perspectives (lines 130-137).
- Line 134-144: It should be made clear that this is opinion. Something like: “The opinion is offered here that it is time to clearly recognize …….”
Reply:
This is our opinion. We have followed your advice (line 137-146).
- Line 145-158: Interesting points are made here that do deserve thought. But placed here, they divert from the line of reasoning in this paper. Furthermore, the authors do not return to these themes in their analyses or in their discussion. In this paper, this portion strikes me as non sequiturs. I recommend deleting Lines 145-158 as not relevant to the present paper.
Reply:
We agree that the parts you mentioned appear as non sequiturs. However, we wanted to convey the meaning that inequality exists between humans and animals even in a welfare state. Furthermore, because it shows the social “hierarchy” and “dominance” relationship that explains the differences in animal welfare policy types in the conclusion, we, after careful consideration, decided to retain the mentioned lines (lines 147-161).
- Line 179-180: A reference is needed here.
Reply:
This position is informed by our research on social welfare policy and field experience (lines 182-183).
- Line :219-227: This paragraph belongs in the Introduction, not in Methods.
Reply:
Thank you for your comment. This aim of this paragraph was to describe the research design (lines 223-229).
- It is clearly designed to evaluate the practices in South Korea in comparison to other nations, and I think it would be better if the title reflected that. I offer something like the following as perhaps more directly descriptive: The National Animal Welfare Policies of South Korea in Comparison to the Legal Frameworks and Systems in Other Countries.
Reply:
We have revised the title as advised for consistency and conciseness. Thank you
Reviewer 3 Report
Comments and Suggestions for Authors
The article entitled "A Comparative Study of National Animal Welfare Policies: Legal Frameworks and Systems across Seven Countries" (Manuscript ID: animals-3515238) provides a comparative analysis of animal welfare policies in seven countries, with the aim of identifying the current status and direction of animal welfare policies in South Korea. The topic addressed offers significant contributions to advancing discussions on the current state of animal welfare policies, with a particular focus on South Korea.
In my opinion, the study is well-planned and technically solid, with great potential for high-quality output. Furthermore, the authors present a relevant study for the scope of the Journal Animals, with a pertinent topic and high citation and readership potential. However, some improvements are necessary to enhance the clarity and flow of the text. I am attaching a file with specific details and comments on the manuscript.
First, the authors should review the manuscript to simplify complex sentences and ensure that each concept flows logically to the next. In some sections, especially in the Results, the content resembles more of a discussion or a detailed description of the methodological process, which could confuse the reader. I recommend that the authors consider revising or removing sentences that do not fit appropriately in the Results section.
Additionally, the authors need to be more direct and clear in their ideas, as the reading becomes difficult at times. The introduction presents the central objective of the study in the first few paragraphs, which is not a common approach. The presentation of the objective could be more concise, avoiding repetition and redundancy throughout the text. I suggest that the authors clarify the objective of the study in a more straightforward way at the end of the introduction. Furthermore, sections 1.1 Theoretical Background and 1.2 The Concept of Animal Welfare Policy and Distinctiveness from Previous Studies on Animals provide a useful theoretical basis, but I believe the essential information, particularly that accompanied by references, could be synthesized directly in the introduction, eliminating the need for these subsections and making the text more direct and objective.
The Materials and Methods section is well-organized, although some points need further clarification. The authors employ a robust and interesting methodology to achieve their objectives. However, some important elements of the section, such as the fuzzy membership scores, are explained in the Results section, which could be reorganized to ensure better cohesion between the sections.
The Discussion section is, in my opinion, the best part of the article. It is well-written and presents a clear and well-founded analysis. However, I notice that the number of references is limited (only 27 throughout the document). Including more relevant works could further strengthen the arguments and conclusions presented. The conclusion is solid and coherent with the study’s results. However, the authors need to make the objective clearer and directly connect it with the conclusion, explicitly responding to the proposed objectives.

Author Response
Response to Reviewer’s Comments
Dear Reviewer #3,
We sincerely thank you for your thoughtful suggestions and insights that have enriched the manuscript and produced a better and more balanced account of the research.
3 Reviewer(s)' Comments to Author:
- Request for revision of "Simple Summary"
Reply:
Thank you for your comments. The "Simple Summary" paragraph has a word limit, so we kept it brief. The "Abstract" has a bit more detail, so we didn't edit it much.
- Request for revision of "Abstract"
Reply:
We have made corrections as per your suggestion, and sentences have been connected systematically.
- Request for revision and supplementation of " Introduction"
Reply:
Lines 34–43 are located at the very beginning because the text is structured in a deductive manner.
The sentences in question express our conclusions based on the studies discussed in Section 1.1.1, “Human Welfare and Animal Welfare.”
The phrase you requested for revision as "in June 2000" is not included because it refers to an approximate, not an exact, period (line 54).
Lines 45–49 indicate that animal welfare is used as an official administrative term, and lines 64–65 indicate an academic debate on the concepts of "animal welfare" and "animal rights," so they were not deleted.
Lines 83-86 are sentences that explain the legitimacy of the welfare policy typology, so they were not deleted.
You are correct that lines 100–110 are an explanation of the research method, but another reviewer requested that the reason for selecting 7 countries should be included; thus, so they were retained to aid readers’ understanding.
- Request for revision of "2.1. Research Model"
Reply:
We decided to leave lines 223-229 as they were after others asked for clarification and merged the paragraphs corresponding to line 235.
- Request for revision of "2.3.1. Fuzzy-Set Ideal Type Analysis(FS/ITA)"
Reply:
We have corrected the abbreviation.
- Request for revision of “2.3.2. Selection of Indicators and Formation of the Property Space”: Clarify and summarize this sentence. Does the analysis of the social value of animal welfare laws contribute to a broader understanding of animal life
Reply:
You suggest that the elements in lines 306-311 should be expressed clearly and simply, but the content became long because an explanation was required before using the term "social value of the Animal Welfare Act" in the analysis.
If animals are recognized as living beings rather than tools for humans, and this is included as a social value in the Animal Welfare Act, the lives of animals in human society will change positively.
- Request for revision of “2.3.3. Dimensions of Each Indicator”
Reply:
“Indicator” represents a higher classification level and should be retained.
The “points” in Tables 1 and 2 have been removed.
Line 331 was pushed back compared to the original version of the paper. It has now been corrected (line 332).
The erstwhile lines 333–335 have been deleted (lines 333-334).
- Request for revision of “3.1. Derivation of Types”
Reply:
Lines 341–358 describe the calculation process that produces Table 3, so they cannot be moved to 2.3.1.
If Tables 3 and 4 were converted to text, they would be excessively long because they would list all the figures, which could cause confusion in understanding the content.
We revised it to be a continuous text instead of a summary like lines 376–378.
Since the reviewer did not suggest where lines 380–382 should be moved to, we left them there to aid understanding of Table 5.
- Request for revision of “3.2.1. Economic Value Type”
Reply:
We agree with the comments on lines 419–426. However, since the fuzzy-set ideal analysis results contain qualitative analysis content, they are sometimes interpreted as the reviewer's opinion.
Reviewer 4 Report
Comments and Suggestions for Authors
This paper adds an interesting dimension to animal law research by mapping animal welfare policies to human welfare policies with the various “types” identified as “economic value”, “social value”, “rights expansion” and “ecological”. However, I found it very difficult to understand the section on the research model and analysis methods and specifically how the criteria were selected and how the typology scores were assigned and evaluated. This may be a function of my own ignorance of the typology analysis used as I am unfamiliar with the FS/ITA methodology. But I imagine anyone not familiar with this typology and methodology will be lost.
The historical discussion at the opening I assume is referring to that of South Korea, because it suggests that animal laws have only gained momentum since 2000 and this is not the case in several other countries. Moreover, it is unclear how the researchers define animal welfare laws versus animal welfare systems (first briefly defined at line 311) and what it counts as animal welfare. This is especially true where they refer to “mere protection” at line 224 which suggests that they are not including anti-cruelty laws for example. Better definition generally of the terms used throughout would be helpful. The researchers note the absence of standardized legal terminology at the conclusion of their paper; thus the need for them to define their own terminology to avoid reader confusion.
Other than the brief list of classifications under Tables 1 and 2 there is very little detailed discussion of the animal welfare laws or systems of the countries studied. The discussion of the types mention the macro approach of each type to animal welfare laws and systems but with very little detail as to the specific laws and systems in place. Further, while I agree with the placement of the US under the “economic value” type, the fact that the analysis reviewed only federal laws, the researchers missed the vast majority of laws governing animals in the US as this is the function of the states not the federal government in the US.
Finally, the discussion section is very short, with the implications and recommendations for South Korea even shorter leaving the reader with only very little concrete suggestions for moving animal welfare laws and systems forward. It would be helpful to expand this discussion some to provide more specific details as to how best to promote animal welfare laws and systems within South Korea if that is in fact an important goal of the piece.
Author Response
Response to Reviewer’s Comments
Dear Reviewer #4,
We sincerely thank you for your thoughtful suggestions and insights that have enriched the manuscript and produced a better and more balanced account of the research.
4 Reviewer(s)' Comments to Author:
- I found it very difficult to understand the section on the research model and analysis methods and specifically how the criteria were selected and how the typology scores were assigned and evaluated.
Reply:
Thank you for your advice. Research model, analysis method, and type score evaluation require separate chapters. It is very difficult to easily and concisely modify the content you requested. We will try to include supplementary explanations in the future research paper.
- It is unclear how the researchers define animal welfare laws versus animal welfare systems (first briefly defined at line 311) and what it counts as animal welfare. This is especially true where they refer to “mere protection” at line 224 which suggests that they are not including anti-cruelty laws for example. Better definition generally of the terms used throughout would be helpful. The researchers note the absence of standardized legal terminology at the conclusion of their paper; thus the need for them to define their own terminology to avoid reader confusion.
Reply:
We appreciate your careful reading of our manuscript. There is a brief explanation of the social value of the Animal Welfare Act and the animal welfare system in lines 219–226, but I agree that an explanation of terms is needed overall. Word count limits prevented much from being included.
- Other than the brief list of classifications under Tables 1 and 2 there is very little detailed discussion of the animal welfare laws or systems of the countries studied. The discussion of the types mention the macro approach of each type to animal welfare laws and systems but with very little detail as to the specific laws and systems in place.
Reply:
We understand your point. However, due to limitations in the length of the paper, detailed information on the countries listed in Tables 1 and 2 was not provided. This is an unedited Korean translation of over 200 pages, and some of the content will be reconstructed for further study, so we apologize for not being able to provide the file.
- While I agree with the placement of the US under the “economic value” type, the fact that the analysis reviewed only federal laws, the researchers missed the vast majority of laws governing animals in the US as this is the function of the states not the federal government in the US.
Reply:
Thank you for your comment. When analyzing the target countries, the same analysis criteria must be applied and a standardized score must be obtained by inputting it into the program. Accordingly, high-level US state laws were excluded.
- The discussion section is very short, with the implications and recommendations for South Korea even shorter leaving the reader with only very little concrete suggestions for moving animal welfare laws and systems forward. It would be helpful to expand this discussion some to provide more specific details as to how best to promote animal welfare laws and systems within South Korea if that is in fact an important goal of the piece.
Reply:
I agree with your comment. This study focused on whether the animal welfare policies of welfare states are similar to the welfare state ideology, whether animal welfare policies can be categorized, and what Korea’s type is. In our next study, we will discuss Korea's animal welfare laws and systems in depth through time-series analysis.
Round 2
Reviewer 2 Report
Comments and Suggestions for Authors
Upon review of the revised manuscript, I am happy to report that my previous concerns have been addressed. I congratulate the authors for producing this fine piece of scholarship.
Reviewer 3 Report
Comments and Suggestions for Authors
The authors have adequately addressed the reviewer’s comments and significantly improved the manuscript. The study addresses a highly relevant and timely topic, fully aligned with the scope and objectives of the journal. The manuscript is clearly structured, methodologically robust, and the results are well-presented and appropriately discussed. The text is well-written, contributing positively to the scientific literature in the field. Therefore, I strongly recommend its publication in its current form.
Reviewer 4 Report
Comments and Suggestions for Authors
The authors did very little, if anything, to address my comments. They merely explained why they were not able to do so. However, I did not consider my comments fatal to the publication of their paper and thus if you wish to proceed to publish, that is ok with me, especially if other reviewers have approved the manuscript.